# Exploring Sociodemographic Correlates of Suicide Stigma in Australia: Baseline Cross-Sectional Survey Findings from the Life-Span Suicide Prevention Trial Studies

**DOI:** 10.3390/ijerph20032610

**Published:** 2023-01-31

**Authors:** Lisa N. Sharwood, Alison L. Calear, Philip J. Batterham, Michelle Torok, Lauren McGillivray, Demee Rheinberger, Stephanie Zeritis, Tuguy Esgin, Fiona Shand

**Affiliations:** 1Black Dog Institute, Faculty of Medicine and Health, University of New South Wales, Sydney 2032, Australia; 2John Walsh Centre, Sydney Medical School, Faculty of Medicine and Health, University of Sydney, St Leonards, Sydney 2006, Australia; 3School of Engineering and Mechatronics, University of Technology Sydney, Broadway, Sydney 2007, Australia; 4Centre for Mental Health Research, iResearch School of Population Health, Australian National University, Canberra 2601, Australia; 5Discipline of Exercise, Health and Performance, University of Sydney, Sydney 2006, Australia; 6School of Medical and Health Sciences, Edith Cowan University, Perth 6027, Australia

**Keywords:** suicide, stigma, Indigenous, help-seeking, prevention, gender

## Abstract

The risk of suicidal behaviour in Australia varies by age, sex, sexual preference and Indigenous status. Suicide stigma is known to affect suicide rates and help-seeking for suicidal crises. The aim of this study was to investigate the sociodemographic correlates of suicide stigma to assist in prevention efforts. We surveyed community members and individuals who had attended specific emergency departments for suicidal crisis. The respondents were part of a large-scale suicide prevention trial in New South Wales, Australia. The data collected included demographic characteristics, measures of help-seeking and suicide stigma. The linear regression analyses conducted sought to identify the factors associated with suicide stigma. The 5426 participants were predominantly female (71.4%) with a mean (SD) age of 41.7 (14.8) years, and 3.9% were Indigenous. Around one-third of participants reported a previous suicide attempt (n = 1690, 31.5%) with two-thirds (n = 3545, 65.3%) seeking help for suicidal crisis in the past year. Higher stigma scores were associated with Indigenous status (β 0.123, 95%CI 0.074–0.172), male sex (β 0.527, 95%CI 0.375–0.626) and regional residence (β 0.079, 95%CI 0.015–0.143). Lower stigma scores were associated with younger age (β −0.002, 95%CI −0.004–−0.001), mental illness (β −0.095, 95%CI −0.139 to −0.050), male bisexuality (β −0.202, 95%CI −0.351 to −0.052) and males who glorified suicide (β −0.075, 95%CI −0.119 to −0.031). These results suggested that suicide stigma differed across the community, varying significantly by sex, sexual orientation and Indigenous status. Targeted educational programs to address suicide stigma could assist in suicide prevention efforts.

## 1. Introduction

Suicide is a critical global public health issue; rates of suicide vary by sex, age, geographic region, cultural background, sexual preference and means. Prevention efforts have been increasing over time, in addition governmental prioritisation of suicide prevention on the national agenda [1,2] via universal or targeted means to address the factors associated with increased risk. The risk factors for suicide death are numerous and can have complex interactions; the stigma of suicide can further impact these risks by perpetuating suicidal thoughts or behaviours [3] and increasing the potential for social exclusion, coercive treatment and contact with the justice system [4]. The stigmatisation of suicide is also deemed to be a central barrier to seeking help and disclosing suicidality (incorporating suicidal ideation and behaviours) [5] for oneself or for others in need of help [1,6]. As a broader effect of suicide stigma inhibiting help-seeking, people may suffer additional preventable physical and mental distress, relationship breakdown, personal stigma and loss of life satisfaction and opportunities [7].

Suicide stigma is defined as the social disapproval or criticism of suicidal thoughts and behaviour and is an additional stressor among persons who experience or have experienced suicidality. This stigma is also experienced by people bereaved by suicide; some describe feeling blame towards themselves as they were unable to assist the person who died [8,9]. Suicide stigma towards oneself is defined as the internalisation of the negative stereotypes and can exacerbate shame and feelings of hopelessness, potentially blocking prevention efforts [3]. Suicidal ideation has been shown to be more prevalent among individuals with high self-stigma [10].

Most of the research investigating perceptions of suicide such as stigma or its converse—glorification—have been conducted in discrete community or clinical populations [11,12,13,14]. However, examining correlates of suicide stigma across different population groups using a representative cross-section of individuals could aid in the further understanding of where similarities or differences lie to better inform prevention activities.

Additional attitudes about suicide that may influence support for individuals experiencing suicidal distress include the glorification or normalisation of suicide. Glorifying suicide has been associated with lower suicide stigma; however, this conversely has a disturbing impact on suicide rates. Hom and colleagues quantified this in a large sample of career firefighters, finding that those who reported a previous suicide attempt during their career were more likely to normalise or glorify it, and this glorification was positively correlated with greater self-reported likelihood of future suicide attempts [15]. 

There are many gaps in our understanding of the impact of culture on suicide stigma, despite broad evidence indicating higher stigma among non-Anglo cultures [16]. Peel et al. [17] identified cultural variations in conceptualisations of suicide with significant differences in willingness to acknowledge it as a concern in some cultures. For example, studies conducted in North America have identified a higher suicide stigma in their Indigenous people compared to their non-Indigenous counterparts [18]. Despite significantly higher suicide rates among Indigenous Australians compared with their non-Indigenous counterparts [19], there is negligible research considering the influence of cultural differences on suicide in Australia, although the impact of colonisation on Indigenous people has been proposed to contribute significantly [20,21]. 

Suicidality is well documented to differ substantially by sex, with higher rates of suicidal ideation among females and higher suicide death rates among males [22]. Sexual identity and preference appears to exert an additional effect, where people identifying as lesbian, gay or bisexual are at an increased risk of lifetime suicide attempts compared to their heterosexual counterparts [23], with some reports suggesting up to a six-fold increased risk [24]. Suicide prevention efforts among sexual minorities that experience discrimination may be further hampered in the presence of suicide stigma [25], and there is a need to better understand the measure of this disparity. 

Despite its importance for suicide prevention, there have been few interventions to reduce suicide stigma in the general population and to support affected persons in dealing with suicide stigma [26]. Further investigation of stigmatising attitudes in communities is needed and in the context of rigorous prevention efforts to understand and document what works. Across New South Wales (NSW), Australia, a population-scale, multi-level intervention called ‘LifeSpan’ [27] has been trialled by the Black Dog Institute, providing this opportunity. 

The LifeSpan trial expressed numerous aims, including to: Increase awareness of suicide-related knowledge and decrease stigma towards suicide at the community-level and among relevant health professionals;Promote help-seeking behaviours among those at risk;Improve suicide literacy and stigma, particularly among people in gatekeeper roles (e.g., emergency services, teachers).

The aim of this study, therefore, was to investigate the association between the key socio-demographic factors that may increase a person’s expression of suicide stigma and to understand if self-reported levels of suicide stigma differed between people who have attempted suicide and general community members prior to the LifeSpan suicide prevention intervention.

## 2. Materials and Methods

This study comprised baseline cross-sectional survey data collected as part of the LifeSpan suicide prevention trial, which was previously described in detail [27]. In brief, the LifeSpan intervention is an integrated suicide prevention framework consisting of nine evidence-based strategies, comprising universal, selective, and indicated interventions, which was implemented as a research trial across four distinct geographic regions in NSW, Australia, during a study period from April 2017–March 2020. To establish a baseline representation of the various sectors of the community and health service, surveys were conducted within selected groups of trial region populations and in various settings. The surveys were set up to capture whether there were changes in the attitudes, awareness and experiences of community and emergency department (ED) attendees over time due to potential exposure to the LifeSpan intervention.

### 2.1. Survey Populations and Recruitment

Survey participants represented members of the general community (“community survey”) and individuals who had attended EDs for suicidal crisis (“RESTORE”). The community survey was initiated to obtain demographic information, measures of suicidal ideation and behaviours and help-seeking from a sample across all trial sites. Prior to the implementation of any LifeSpan intervention within these regions, baseline data were obtained to assess beliefs, attitudes and knowledge around suicide. Subsequent community surveys were taken at various time points during and post the trial implementation; however, the data used in the current study represented only the baseline timepoint. Eligible community survey study participants were self-identified general community members, recruited predominantly online through paid Facebook advertisements. The survey was commenced following the provision of a study information sheet (online), whereby all persons gave their informed consent prior to their inclusion in the study; respondents were not reimbursed for their participation.

The RESTORE survey used a mixed-methods prospective cohort design with data collected from LifeSpan trial sites and control sites, which was detailed in the published study protocol [28]. Eligible participants were individuals who had presented to an ED following a suicidal crisis in the prior 18 and 12 months for cohort one and two, respectively, and were recruited online through paid Facebook advertisements or via ED staff in participating trial sites who handed out study information flyers to eligible participants. The key data collected across both the community and RESTORE surveys formed the basis for the correlates of stigma included in the current study. 

### 2.2. Survey Instruments and Variables

Consenting participants responded to a series of demographic questions including age, sex (male, female, non-binary), Indigenous status (Aboriginal, Torres Strait Islander, both, neither), marital status (married, de facto, single, divorced, separated, widowed) employment status (full-time, part-time, unemployed), residence (metropolitan, inner regional, outer regional/remote), sexual orientation (heterosexual, homosexual, bisexual, other), self-reported history of mental illness (Yes/No) or suicide attempt (Yes/No) and whether they had sought help in the past year for suicidality (Yes/No). The ‘Stigma of Suicide Scale’ (SOSS) [29] was included in both surveys to assess attitudes towards people who die by suicide. The SOSS shows a three-factor structure, which has been replicated in multiple samples internationally [29,30,31]. The sixteen-item short form of the SOSS was used in the present study, which included eight items assessing suicide stigma, four items assessing glorification/normalisation of suicide and four items attributing suicide to depression or isolation. Each item consisted of a one- or two-word descriptor of a person who dies by suicide, rated on a five-point Likert scale from (1) strongly disagree to (5) strongly agree. The subscales of the SOSS were calculated by obtaining the mean response to all items on the subscale, ranging from 1 to 5. The SOSS subscales have previously demonstrated robust internal consistency [31]. The SOSS stigma subscale was used as the outcome measure for this study as a continuous numeric variable. The Strengthening the Reporting of Observational Studies in Epidemiology (STROBE) statement [32] was used to report the study management and findings appropriately. 

### 2.3. Data Management and Assumptions

Data from the two surveys were appended after first cleaning and ensuring all variables were coded alike. Very few respondents categorised themselves as non-binary and were therefore also removed from the final dataset. Missingness was examined, and the median was imputed if missing at random for ≤5% for continuous variables (excluding any of the three SOSS subscales). Participants from the original survey populations who did not respond to any of the SOSS questions were excluded from the analysis. As the SOSS stigma subscale was used as the outcome of interest, respondents with less than 7 of the 8 subscale responses were excluded. 

### 2.4. Statistical Analysis

All survey responses were entered into purpose-built Qualtrics survey databases and then downloaded into Microsoft Excel (Microsoft Corporation, Washington). Analysis for this study was conducted using STATAv16 [33] and in RStudio [34], with the figures being produced using the package ggplot2 [35]. Variable reporting followed internationally standardised formats; parametric data were described using mean and standard deviations. *p*-values of association were significant at <0.05. Predictors were assessed for multicollinearity using the ‘cor’ command. An ordinary least-squares regression model was fitted using the mean of the stigma subscale of the SOSS as the response variable [29] and other variables of interest or significance as the predictors. Analysis was commenced by fitting a model with the mean of the SOSS score as the response and other variables of interest or significance as the predictors. The first full model comprised all predictors, and the second included the interaction terms of significance based on the initial exploratory data analysis in addition to all interactions with the sex variable. Model selection was conducted using the ‘step’ function in the R base package [36], which was reduced by comparing all possible models sequentially by evaluation of Akaike’s information criterion (AIC). Variables removed throughout this stepwise process included marital status, interactions between sex and help-seeking, as well as mental illness diagnoses and employment status. The most parsimonious model was selected as that with the lowest AIC. Examination of the residuals was undertaken by creating a residual plot, plotting the least-squares residuals against ŷ. We assessed for the presence of heteroskedasticity for conducting the Breusch–Pagan test. The lmtest package [37] and bptest function were used to evaluate our fitted model.

## 3. Results

Eligible respondents totalled 5426 individuals across the two combined surveys; 4283 (78.9%) from the community survey and 1143 (21.1%) from the RESTORE survey. The total respondents were predominantly female (71.4%) with a mean (SD) age of 41.7 (14.8) years, and 3.9% were identified as Indigenous. Around one third of the participants reported a previous suicide attempt (n = 1690, 31.5%), and almost two-thirds of the study population (n = 3545, 65.3%) had sought help for suicidality in the past year. Table 1 describes the characteristics of the study population, presented by survey group and total. 

ED attendees for suicidality from the RESTORE study were more likely to describe themselves as homosexual (8.3%) or bisexual (21.3%) than individuals from the community survey (4.5% and 6.8%, respectively). ED attendees were also more likely to report being single (60.6%) and unemployed (42.5%) than the community survey participants (20.8% and 26%, respectively). 

Further investigation of the summary values (mean [SD]) of the SOSS subscales showed distinctly higher stigma values among males, Indigenous persons and heterosexual participants (Table 2). Glorification scores were highest among homosexual people, and scores for attribution to isolation were highest among bisexual people and Indigenous people.

Figure 1 shows a boxplot which depicts the differences in the distributions of the SOSS stigma subscale mean scores for males compared with females and across each category of sexual orientation, expanding on that shown in Table 2. 

After assessing for multicollinearity and finding −0.05 to be the highest value, we determined there was no evidence of multicollinearity. All the covariates retained in the most parsimonious model (selected as that with the lowest AIC) and their effect estimates are shown in Table 3. Male sex was associated with the greatest magnitude of effect increasing the suicide stigma mean scores (compared to female sex) (β 0.527, 95%CI 0.375 to 0.626); however, higher glorification of suicide scores (SOSS subscale) were a substantial effect modifier for this (i.e., glorification had a larger negative impact on stigma for males than females), which was retained in the most parsimonious model as an interaction (β −0.075, 95%CI −0.119 to −0.030). To establish the veracity of this linear regression model, we performed a residuals analysis using the plot function. This showed a linear pattern, with the residuals in the qq-plot appearing normally distributed. The residual standard error was 0.639 on 5404 degrees of freedom. The Breusch–Pagen test for heteroskedasticity (LM-BP Test) result was 14.21. Therefore, the null hypothesis was rejected at the 5% level (χ2 0.05 ≈ 3.87, *p* = 0.0001). 

Indigenous status (compared to non-Indigenous status) was associated with a higher suicide stigma (β 0.123, 95%CI 0.074 to 0.172). Area of residence was also found to influence the suicide stigma scores, where higher scores were associated with outer regional or remote areas of residence compared with metropolitan areas (β 0.079, 95%CI 0.015 to 0.143). Finally, and to a lesser degree, the isolation subscale scores were associated with higher stigma of suicide scores (β 0.040, 95%CI 0.017 to 0.063).

Younger ages (continuous variable) demonstrated lower stigma scores (β −0.002, 95%CI −0.004 to −0.001), and sexual preference was also significantly associated with a lower stigma of suicide. Lower stigma scores were associated with homosexual and bisexual respondents compared to heterosexual respondents (β −0.127, 95%CI −0.182 to −0.021 and β −0.056, 95%CI −0.135 to −0.032 respectively). Male bisexuality was again associated with lower stigma scores (β −0.201, 95%CI −0.350 to −0.052).

Differences in the reported stigma of suicide were clear between the survey populations and were retained as significant in the final model, whereby participants attending an ED for suicidal crisis (RESTORE survey) had significantly higher stigma scores than participants in the community survey (β 0.08, 95%CI 0.036 to 0.132). The adjusted R^2^ in the final model was 0.17. 

The mean scores for the SOSS subscales of suicide stigma and glorification differed by sex and Indigenous status. The stigma scores among Indigenous males contrasted significantly to Indigenous females, as demonstrated clearly in Figure 2, whose plot was produced using the ggplot2 package in R. Indigenous male participants had greater reductions in stigma with a higher glorification compared to non-Indigenous males, where there were no differences in the relationship between Indigenous and non-Indigenous females.

## 4. Discussion

This study involved a large sample of two populations, where self-reported stigma of suicide was identified to be highest among Indigenous people, males, heterosexual persons and those living in outer regional or remote areas of NSW during the study period, 2017–2020. These findings contribute to and should be considered in the context of the developing evidence bases regarding Indigenous suicide research internationally, which focusses on holistic, culturally safe and strength-based approaches that promote social and emotional wellbeing.

Suicide rates among Indigenous Australians are twice that of the non-Indigenous population [22]. The factors identified with heightened suicide among Indigenous Australians include challenges surrounding acculturation and minority group status, discrimination, socioeconomic deprivation, poverty, unemployment, and inequalities of access to healthcare and service provision [38]. A greater understanding of the role of suicide stigma in pathways to Indigenous suicide can be used to encourage and foster resilience, as discussed by Dudgeon et al. [20,21].

No representative Australian data are available to reliably determine the rates of suicide and suicide attempts amongst the LGBTQI+ community, although the international literature indicates that minority sexual orientation is associated with an increased prevalence of suicidal behaviour [39]. 

This study found that groups of people typically characterised as having a high suicide risk (Indigenous people and men) also had high suicide stigma scores, whereas stigma scores were comparatively low among homosexual and bisexual people, despite their documented levels of suicidality compared with heterosexual people [23,24]. The high suicide stigma in Indigenous populations may have developed alongside the historical mistrust of Western systems, including healthcare systems, which have largely ignored the importance of practicing, acknowledging and reclaiming culture as vital to Indigenous good health and wellbeing [19,40,41]. The Australian Government created the National Aboriginal and Torres Strait Islander Suicide Prevention Strategy in 2013 [42]; however, this did not specifically address stigma, and as such more needs to be done at the government and community levels to explore and address the links between the historical trauma associated with colonisation and the stigma of suicide among Indigenous Australians [43]. A recently published scoping review, which thematically examined 72 articles about Indigenous suicide prevention, identified that culturally grounded suicide education and awareness initiatives reduced stigma towards suicide and increased willingness to seek help [44]. Our study did not evaluate any past experiences with suicide prevention interventions. 

The strengths of this study included the incorporation of novel factors associated with stigma, its large, diverse sample of general community and clinical populations, and a validated measure of suicide stigma.

This study had several limitations. While all the surveyed populations were asked a core set of questions that were compiled in one dataset for the purpose of this study, the groups within this study were comprised from two discrete studies and represented somewhat heterogenous and non-representative populations. Some measures were therefore notably higher in one group over the other. For example, individuals in the RESTORE study reported a higher prevalence of mental illness and/or suicide attempt than the other study group. The inclusion of the survey indicator as a factor in the model, however, attempted to account for this as much as possible. To the authors’ knowledge, the stigma of suicide scale (SOSS) has not been evaluated for its cultural appropriateness in Indigenous populations. This further work is needed, as the results may otherwise differ. Finally, as the baseline study was cross-sectional, the direction of effects could not be established. 

The implications from the findings in this study include recommendations for a focus on education programs or campaigns [45,46,47,48], which have been previously researched and shown as effective. Other potentially effective approaches might include:

(1) Changes to health system policies and service pathways to better support individuals experiencing suicidal distress;

(2) Contact interventions to challenge stereotypes and promote messages related to recovery [46]; 

(3) De-stigmatisation through media representation of suicide [49];

(4) Peer services to destigmatise the treatment of suicidal distress; 

(5) The development of alternatives to ED that are more person-centred, less stigmatising and with less potential for coercive control.

## 5. Conclusions

The results of this study suggested that suicide stigma was diverse across the population, varying by sociodemographic factors. Higher levels of suicide stigma were found among people identifying as Indigenous, male, and heterosexual after controlling for other relevant variables. This provided new information as to the higher-risk subgroups, within which there was also higher stigma. These identified subgroups of the population could benefit from targeted educational programs to address what is likely a higher risk of stigmatisation toward their peers. Campaign messages should focus on suicidality as a complex phenomenon with multiple interactive causes, many of which can be treated, supported, and addressed through multisystem and culturally sensitive approaches to suicide prevention. 

## Figures and Tables

**Figure 1 ijerph-20-02610-f001:**
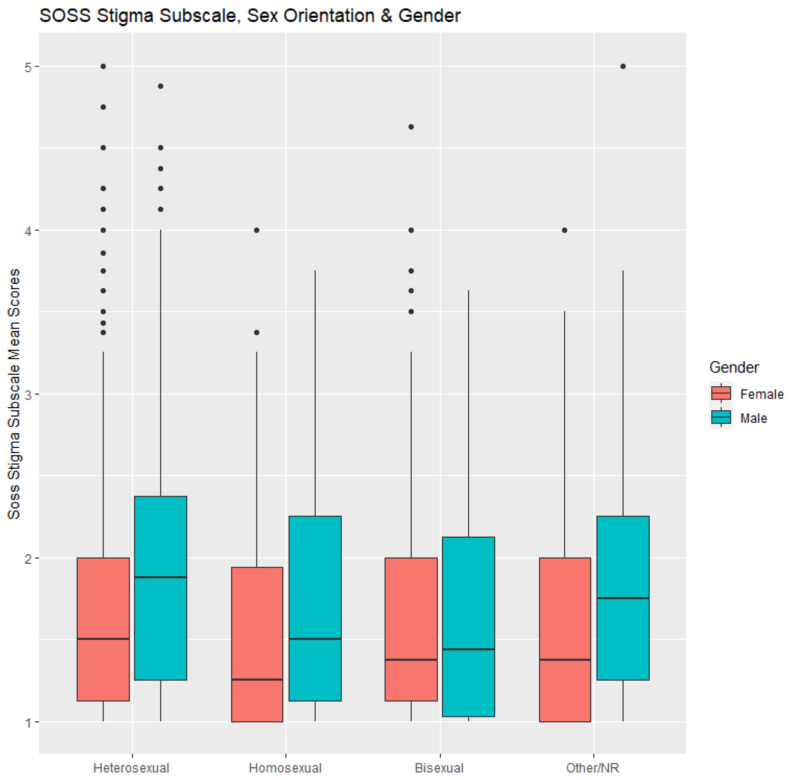
Boxplot of mean stigma scores by sexual orientation and sex.

**Figure 2 ijerph-20-02610-f002:**
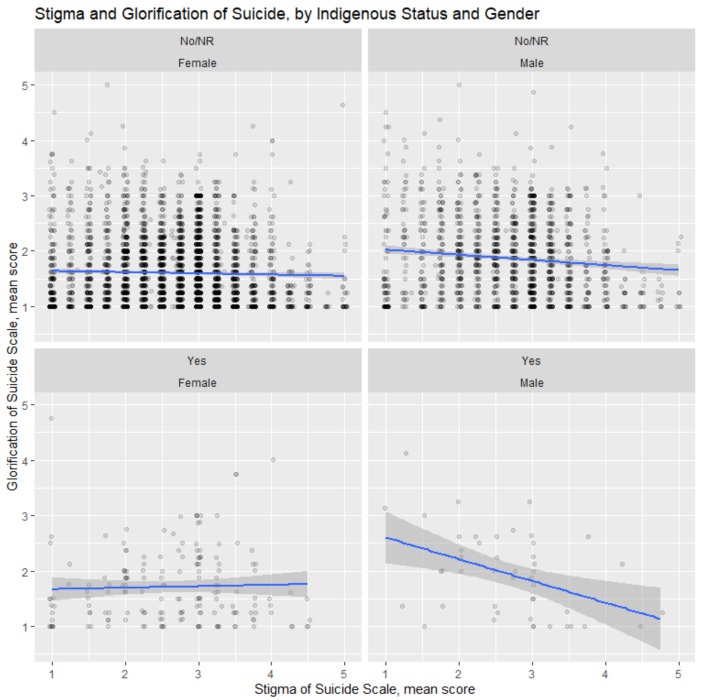
Visualisation of mean stigma and mean glorification scales faceted by sex and Indigenous status.

**Table 1 ijerph-20-02610-t001:** Characteristics of the study population.

Characteristic	Community Survey N = 4283	RESTORE * N = 1143	TOTAL N = 5426
Age (mean [SD])	44.5 (13.9)	29.7 (11.9)	41.4 (14.8)
Sex			
- male	1369 (31.9)	184 (16.1)	1553 (28.6)
- female	2914 (68.0)	959 (83.9)	3873 (71.4)
Sexual orientation			
- heterosexual	3399 (79.3)	705 (61.7)	4104 (75.6)
- homosexual	193 (4.5)	95 (8.3)	288 (5.31)
- bisexual	294 (6.8)	244 (21.3)	538 (9.9)
- other/not recorded	397 (9.3)	99 (8.7)	496 (9.1)
Indigenous (yes)	137 (3.2)	79 (6.9)	216 (3.9)
History of mental illness			
- yes	2864 (66.9)	1102 (96.4)	3966 (73.1)
- no	1419 (33.1)	41 (3.6)	1460 (26.9)
Previous suicide attempt			
- yes	1077 (25.1)	613 (53.6)	1690 (31.1)
- no	3206 (74.8)	530 (46.3)	3736 (68.8)
Sought help past year for suicidality			
- yes	2453 (57.3)	1092 (95.5)	1881 (34.6)
- no	1830 (42.7)	51 (4.5)	3545 (65.3)
Marital status			
- single	891(20.8)	693 (60.6)	1584 (29.2)
- married/de facto/	2326 (54.3)	347 (30.4)	2673 (49.3)
- divorced/separated/widowed	753 (17.6)	103 (9.0)	856 (15.8)
- not reported	313 (7.3)	0 (0)	313 (5.7)
Employment status			
- full-time	1765 (41.2)	286 (25.0)	2051 (37.8)
- part-time	1091 (25.5)	371 (32.5)	1462 (26.9)
- unemployed	1114 (26.0)	486 (42.5)	1600 (29.5)
- not reported	313 (7.31)	0 (0)	313 (5.7)
Residence			
- metropolitan	2750 (64.2)	750 (65.6)	3500 (64.5)
- inner regional	1204 (28.1)	289 (25.3)	1493 (27.5)
- outer regional/remote	329 (7.7)	104 (9.1)	433 (7.9)

* RESTORE = Recording Experiences of Suicidality TO Reform Emergency care.

**Table 2 ijerph-20-02610-t002:** Stigma of suicide scale summary scores across demographic factors.

Characteristic	Stigma Subscale Mean (SD)	Glorification Subscale Mean (SD)	Isolation Subscale Mean (SD)
Sex			
- male	1.87 (0.71)	2.64 (0.84)	3.96 (0.76)
- female	1.61 (0.62)	2.60 (0.85)	4.01 (0.75)
Indigenous status			
- yes	1.78 (0.69)	2.63 (0.87)	4.06 (0.82)
- no	1.67 (0.64)	2.61 (0.84)	3.99 (0.75)
Sexual Orientation			
- heterosexual	1.70 (0.66)	2.59 (0.84)	3.99 (0.75)
- homosexual	1.57 (0.65)	2.84 (0.90)	3.96 (0.78)
- bisexual	1.60 (0.65)	2.66 (0.89)	4.10 (0.72)
- other/not recorded	1.67 (0.64)	2.65 (0.83)	3.95 (0.77)
Marital Status			
- married/de facto	1.69 (0.66)	2.55 (0.83)	3.95 (0.76)
- divorced/separated/widowed	1.71 (0.67)	2.67 (0. 80)	3.99 (0.76)
- single	1.66 (0.65)	2.70 (0.88)	4.10 (0.72)
- not recorded	1.70 (0.62)	2.63 (0.81)	3.92 (0.77)
Age groups			
- 16–30 years	1.67 (0.66)	2.63 (0.89)	4.10 (0.73)
- 31–45 years	1.72 (0.68)	2.63 (0.84)	4.033 (0.75)
- 46–60 years	1.66 (0.63)	2.59 (0.83)	3.95 (0.76)
- 61–75 years	1.70 (0.62)	3.79 (0.76)	2.63 (0.75)
- 75+ years	1.38 (0.35)	2.54 (0.85)	3.56 (0.92)

**Table 3 ijerph-20-02610-t003:** Multivariate linear regression model of factors significantly associated with the SOSS suicide stigma subscale.

Variable	Coefficient	*p*-Value	95%CI
Age	−0.002	<0.001	−0.004 to −0.001
Sex			
- female	(ref)		
- male	0.527	<0.001	0.375 to 0.626
Sexual orientation			
- heterosexual	(ref)		
- homosexual	−0.127	<0.001	−0.182 to −0.021
- bisexual	−0.056	0.01	−0.135 to −0.032
- other/not stated	0.035	0.10	−0.003 to 0.064
Indigenous status			
- no	(ref)		
- yes	0.123	<0.001	0.074 to 0.157
History of mental illness			
- no	(ref)		
- yes	−0.108	<0.001	−0.139 to −0.050
Sought help past year			
- no	(ref)		
- yes	−0.068	<0.001	−0.111 to −0.025
Residence			
- metropolitan	(ref)		
- inner regional	0.022	0.10	−0.010 to 0.067
- outer regional/remote	0.079	<0.001	0.015 to 0.143
SOSS isolation subscale ^#^	0.040	<0.001	0.017 to 0.063
SOSS glorification subscale ^#^	−0.014	0.250	−0.038 to 0.009
Survey			
- community	(ref)		
- RESTORE	0.08	<0.001	0.036 to 0.132
Interactions			
sex(male) ∗ SOSS glorification subscale ^#^	−0.075	<0.001	−0.119 to −0.030
sex(male) ∗ Indigenous (yes)	−0.086	<0.001	−0.112 to −0.061
sex(male) ∗ homosexual	−0.114	0.156	−0.273 to 0.044
sex(male) ∗ bisexual	−0.201	0.002	−0.350 to −0.052
sex(male) ∗ other/not stated	−0.047	0.467	−0.174 to 0.080
Intercept	1.645	<0.001	1.513 to 1.777

^#^ Mean score.

## Data Availability

The data obtained from these surveys may be made available following request to the authors and only after human research ethics approval.

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
