# Peer review of "Exploring Sociodemographic Correlates of Suicide Stigma in Australia: Baseline Cross-Sectional Survey Findings from the Life-Span Suicide Prevention Trial Studies"

_ijerph, 2023, doi:10.3390/ijerph20032610_

Round 1
Reviewer 1 Report
Well written study with a strong literature review and methodology. Large sample size, although heavily weighted to female gender. Results are thorough and graphs help supplement the findings. Overall I felt this was a well-done study and didn't find any problems with it. The biggest challenge I have with the study is that I didn't find it particularly helpful, useful or advancing the knowledge or science in this area. There just was not anything new to me in this study that isn't already known. There is merit in this in that it validates other work, but I felt it lacked anything that advanced anything to help the field.
Author Response
We thank the reviewer for their report and positive evaluation of our manuscript. We would like to highlight the paucity of literature comparing stigma towards suicide by Indigenous persons compared with non Indigenous persons. We believe our study offers a unique finding in this regard. We have also added reference to a recently published review paper and some text in the Discussion (lines 280-84) - to highlight this and describe relevant findings from this review... participants in culturally-grounded suicide education and awareness initiatives demonstrated reduced stigma towards suicide and increased willingness to seek help.
Reviewer 2 Report
Thank you for the opportunity to read your research. As you note, this is a hugely important public health issue. The paper reports the findings of a significant research project. The paper is very well presented and argued. It contextualises the project in the wider context demonstrating a clear understanding of the ethical, therapeutic and other issues in this field.
Author Response
We thank the reviewer for their comments and positive regard for what we believe to be an important study.
Reviewer 3 Report
Line 172: add a citation for ggplot2. Similar issues to function ``step''. In particular, the R version, function version, and package version shall be specified.
There is no residual analysis for the regression models in the manuscript. If any of the assumptions are violated, that will bring immediate impact on Table 3 and conclusions from it.
The residual analysis must be included to make the manuscript publishable.
Author Response
We thank the reviewer for their time and attention to the detail of our referencing and statistical output in particular. We have among our authorship team, two Professors of Statistics (Batterham and Calear), which may reassure you. In the model building process we did indeed run the residulas plot on the model selected for best fit as described... (plot(lmod1f), and evaluated qqnorm, qqline as well as the log of these. We have added the output of this to the manuscript as requested.
Regarding the reviewer's 'check box' notations, we believe the research design is appropriate to address the topic of interest, however do make note of the limitations recognised.
We have added additional references for the base R step function, and the ggplot package used, within the statistical analysis section as requested.
Round 2
Reviewer 3 Report
The revision clears up the normality of errors for the regression model. However, I did not see the examination of heteroscedasticity in the residuals. Please also confirm the homoscedasticity of the residuals.
It is also great to include the procedure of residual analysis in the appendix to clear up readers' concerns.
Author Response
We sincerely thank the reviewer for their diligence to the statistical aspects of this model, and apologise for not adding the specifics of the tests for heteroskedasticity. We did include this in the evaluation of the model fit, and have added further information to the text regarding the Breusch-Pagan test for heteroskedasticity in the text (for ease to to the reader as opposed to an appendix). We hope this clarifies any outstanding questions the reviewer may have. Regards, Dr Lisa Sharwood